**Data Availability Statement:** The code and data used in the model are in GitHub at: https://github.

# An evidence synthesis approach for combining different data sources illustrated using entomological efficacy of insecticides for indoor residual spraying

Nathan Green[1]*, Fiacre Agossa[2], Boulais Yovogan[2], Richard Oxborough[3], Jovin Kitau[4], Pie Müller[5,6], Edi Constant[7], Mark Rowland[8], Emile F. S. Tchacaya[7], Koudou G. Benjamin[7], Thomas S. Churcher[9], Michael Betancourt[10], Ellie Sherrard-Smith[9]

**1** Department of Statistical Science, University College London, London, United Kingdom, **2** Centre de Recherche Entomologique de Cotonou, Quartier DONATIN/AKPAKPA, Benin, **3** Abt Associates Inc., Rockville, Maryland, United States of America, **4** Department of Parasitology, Kilimanjaro Christian Medical University College, Moshi, Tanzania, **5** Swiss Tropical and Public Health Institute, Basel, Switzerland, **6** University of Basel, Basel, Switzerland, **7** Centre Suisse de Recherches Scientifiquesen Côte d'Ivoire, Abidjan, Côte d'Ivoire, **8** London School of Hygiene and Tropical Medicine (LSHTM), London, United Kingdom, **9** Faculty of Medicine, MRC Centre for Global Infectious Disease Analysis, Department of Infectious Disease Epidemiology, School of Public Health, Imperial College London, London, United Kingdom, **10** Symplectomorphic, LLC, New York, NY, United States of America

* n.green@ucl.ac.uk

## Abstract

### Background

Prospective malaria public health interventions are initially tested for entomological impact using standardised experimental hut trials. In some cases, data are collated as aggregated counts of potential outcomes from mosquito feeding attempts given the presence of an insecticidal intervention. Comprehensive data i.e. full breakdowns of probable outcomes of mosquito feeding attempts, are more rarely available. Bayesian evidence synthesis is a framework that explicitly combines data sources to enable the joint estimation of parameters and their uncertainties. The aggregated and comprehensive data can be combined using an evidence synthesis approach to enhance our inference about the potential impact of vector control products across different settings over time.

### Methods

Aggregated and comprehensive data from a meta-analysis of the impact of Pirimiphos-methyl, an indoor residual spray (IRS) product active ingredient, used on wall surfaces to kill mosquitoes and reduce malaria transmission, were analysed using a series of statistical models to understand the benefits and limitations of each.

### Results

Many more data are available in aggregated format ($N$ = 23 datasets, 4 studies) relative to comprehensive format ($N$ = 2 datasets, 1 study). The evidence synthesis model had the

com/n8thangreen/malaria_evidence_synthesis/
tree/plos-one.

**Funding:** The Medical Research Council provided
financial support for ES-S [MR/T041986/1]. The
funders had no role in study design, data collection
and analysis, decision to publish, or preparation of
the manuscript. The specific roles of these authors
are articulated in the author contributions section.
The remaining authors received no specific funding
for this work."

**Competing interests:** The authors have declared
that no competing interests exist.

smallest uncertainty at predicting the probability of mosquitoes dying or surviving and blood-feeding. Generating odds ratios from the correlated Bernoulli random sample indicates that when mortality and blood-feeding are positively correlated, as exhibited in our data, the number of successfully fed mosquitoes will be under-estimated. Analysis of either dataset alone is problematic because aggregated data require an assumption of independence and there are few and variable data in the comprehensive format.

## Conclusions

We developed an approach to combine sources from trials to maximise the inference that can be made from such data and that is applicable to other systems. Bayesian evidence synthesis enables inference from multiple datasets simultaneously to give a more informative result and highlight conflicts between sources. Advantages and limitations of these models are discussed.

## Introduction

Interventions that shorten the mean lifespan of a mosquito and interrupt biting cycles are integral to the control of malaria infections across Africa [1]. The entomological effect of the indoor residual spraying of insecticide (IRS) and insecticide-treated nets are tested using experimental hut trials as part of the product validation process and before World Health Organization (WHO) recommendations can be considered and made [2]. These trials apply IRS to huts and then the temporal impact of the IRS is tracked using human study participants who stay overnight to act as bait for blood-seeking mosquitoes. Over the course of the malaria transmission season, when mosquitoes are present, the outcomes of mosquito mortality and feeding attempts are observed and compared to those incurred for a study participant who stays overnight in an unsprayed hut. On entering an IRS treated hut, a mosquito may: i) blood-feed on human-baits; ii) be killed by the IRS chemical compound, or; iii) exit into window or veranda traps.

Data may be recorded in what we shall call *comprehensive* or *aggregate* formats. Aggregate data provide total counts of a particular outcome but do not account for the counts of other outcomes. Alternatively, comprehensive data provide more detailed information about subgroups representing two or more outcomes. These hut trial data are often published in aggregated format as the key metrics outlined by the WHO (induced mortality, blood-feeding inhibition and deterrence) [2] do not need data to be disaggregated. The effects of interventions must be replicated in multiple settings that have different ecological characteristics to better understand the overall protection that IRS–or other vector control–can afford. Systematic reviews are increasingly used to assess ecological trends in these combined data and summarise evidence to help evaluate interventions [3–5].

Compiling aggregated data (AD) such as experimental hut studies can cause complications for meta-analyses that use the data for slightly different purposes because, among other reasons: i) AD can be presented in inconsistent ways by summarising results [6] making data hard to harmonize across different trials [7]; ii) AD may not fully account for characteristics evident in comprehensive data (CD) leading to ecological bias [8,9]; iii) if large numbers of large trials are available, meta-regression analyses of AD may prove statistically powerful but with few or smaller trials AD may miss clinically significant treatment level effects [10], and; iv) within study variability may be missed [11]. However, there are often few data sets available

for any intervention tested and, historically, only a subset of these trials may provide the CD, which could alleviate some of these issues.

In the experimental hut data testing IRS products, a specific challenge arises from aggregating data because no distinction is made as to whether mosquitoes have blood-fed and survived or blood-fed and been killed. This is an important epidemiological distinction because those mosquitoes that blood-feed and survive may go on to oviposit or transmit malaria parasites onward. Our recent assessment of IRS products made the assumption that mosquitoes were equally likely to have blood-fed and survived or blood-fed and died on entering a sprayed hut [3]. However, IRS exploits the resting behaviour of mosquitoes after feeding, so we need a method to capture the likely higher mortality in fed mosquitoes. In this paper, we use systematically collated data from Sherrard-Smith *et al.* [3] on the IRS active 300g/L Pirimiphos-methyl as an example dataset to explore different models that aim to infer how the impact of the vector control product on mosquitoes changes over time using both AD and CD. Within these data we have 4 studies presenting AD (23 time series), of which 1 study also presents CD (2 time series). Careful consideration is required particularly for any analysis where data are aggregated in different ways across trials, and where the comprehensive data are only part of the total available data. Ideally, we want to ensure that any inference using AD is in agreement with inference afforded by the comprehensive data. Bayesian statistical methodologies provide a natural paradigm to analyse evidence from multiple sources in different formats [11–16]. Bayesian evidence synthesis is a framework that explicitly combines data sources enabling joint estimation of parameters and their uncertainties [17]. We compare predictions of each statistical model presented to those estimated by analysing the subset of either aggregated data or comprehensive data individually. We demonstrate the advantage of inferring from both data sources.

## Methods

We apply the models to an empirical dataset to explore how mosquito outcomes can be interpreted from experimental hut trials testing the efficacy of an IRS product.

### Empirical data

Briefly, a meta-analysis of IRS experimental hut trials was previously conducted on metrics of IRS efficacy [3]. PRISMA guidelines were followed to highlight how best to conduct the systematic review. The outcome metrics of interest are count data for mosquitoes over a time series of multiple months. The original review used four search engines (Web of Knowledge, PubMed, JSTOR and Google Scholar) to find relevant data resources. For the present analysis, these previously published data were then divided into those studies reporting summary data; that is, the total number of mosquitoes, the total number fed, or killed during the trial. These form the aggregated dataset. Other time series included a comprehensive division of mosquito outcomes; that is, the total number of mosquitoes, the number that had fed and died, fed and survived, or not fed and died, or not fed and survived. The aggregated dataset included data without these distinctions, so it is not possible to know whether a fed mosquito was also dead.

The IRS product data used consists of the organophosphate insecticide Pirimiphos-methyl that is widely used for IRS campaigns across the African continent since WHO recommendation in 2013 [18]. The product was evaluated using West African experimental huts in Benin, Côte d'Ivoire and Tanzania across 4 studies [19–22]. Twenty three datasets from these studies had aggregated data reporting the total number of mosquitoes that entered sprayed huts, and the total number killed, blood-fed or exited without the distinction for comprehensive assessment for at least 3 time points (S1 Appendix). Comprehensive data were available from two of

these datasets (S1 Appendix) (data resources are summarised in Table 1). At least 3 repeated measures through time were made for each study, ranging from 6 to 12 months since the insecticide was first deployed.

Our aim is to determine the probability that a mosquito is either killed or blood-fed and surviving after attempting to feed in an experimental hut sprayed with insecticide. We determine how this association changes over time and how probabilities differ between mosquito species.

A contingency table to demonstrate the different aggregation of these data from the available sources is shown in Fig 1.

## Statistical methods

We built three statistical models that were fitted in a Bayesian framework using the probabilistic programming software OpenBUGS (release 3.2.3) [23], accessed through R (version 3.6.1) [24]. Data and code are provided in S2 Appendix. We provide equivalent code for the same process to be conducted in RStan (version 2.21.1) [25] in S2 Appendix. All models are represented schematically in Fig 2.

The first model (in Model 1) uses the aggregated data only and estimates the proportion of mosquitoes that are killed and those that are blood-fed before inferring from the product of these probabilities, those that are surviving and blood-feeding, using the assumption that a mosquito is equally likely to be blood-fed whether killed or not.

The second model adjusts the aggregated data directly prior to fitting the model and then fits a logistic binomial to estimate the respective probabilities from the adjusted data. This is an approach used previously [3] on aggregated data (Model 2a). This model structure is also fitted to the comprehensive data where no prior assumptions are required (Model 2b) to estimate the probabilities from the comprehensive data set.

The third model (Model 3), a Bayesian evidence synthesis model, is constructed similarly but distinct from previous methods [16,17,26]. This model combines the data sources probabilistically to incorporate the inferences that can be made from the comprehensive data benefited by the additional aggregated data source.

**Model 1.** The largest number of available trials provide AD. Model 1 disregards the smaller subset of richer CD. These trials do not record which mosquitoes had jointly survived and blood-fed. Posterior distributions are estimated for the marginal probabilities of mosquitoes that are blood-fed ($P^f$) or mosquitoes that have been killed ($P^d$) (Fig 1).

**Table 1. A list of the studies included and which models are informed by each dataset.** Where indicated, the data are provided in full, in S1 Appendix.

| Data source | Aggregated data | Comprehensive data |
| --- | --- | --- |
| [21] (using pirimiphos methyl 2 gm$^{-2}$) | 1 time series | - |
| [19] (using pirimiphos methyl B 30% CS, pirimiphos methyl AA 30% CS, testing *An. gambiae* sl and *An funestus* sl) | 16 time series | - |
| [22] (using pirimiphos methyl B 30% CS, pirimiphos methyl BM 30% CS, mud walls, *An arabiensis*) | 2 times series | - |
| [20] (using pirimiphos methyl B 30% CS, mud and cement walls, testing *An gambiae* sl and *An. funestus* sl) | 4 time series | 2 time series (*Study 1 cement, 0.5 gm$^{-1}$) (*Study 2 cement, 1 gm$^{-1}$) |
| Total | 23 | 2 |

*reference for Fig 6.

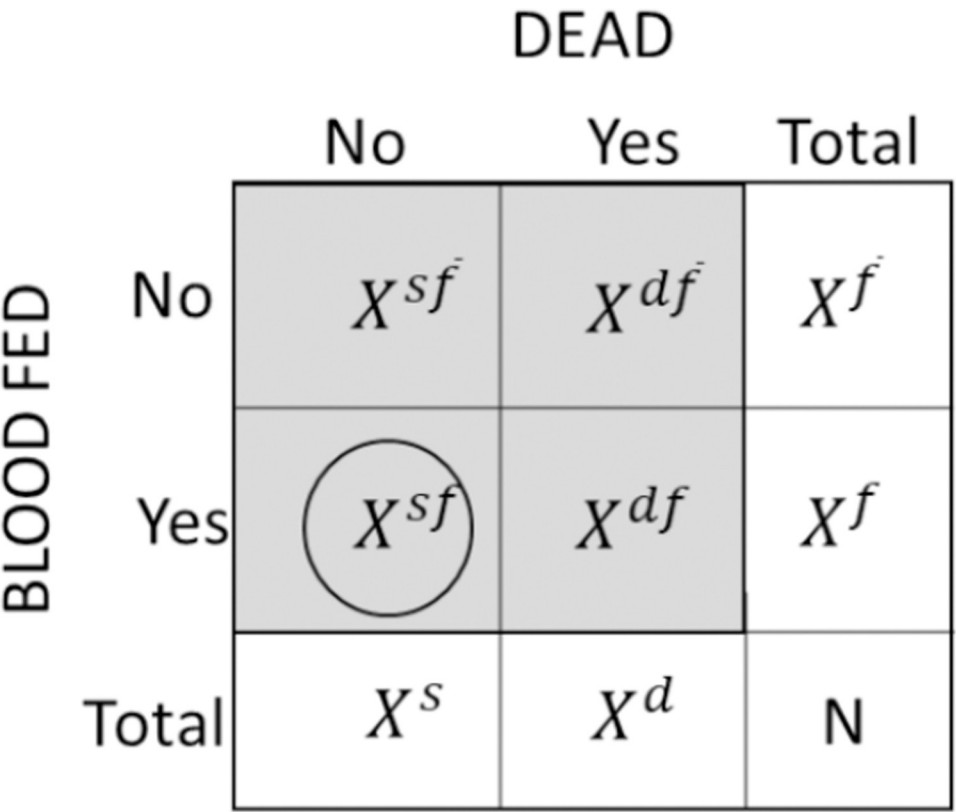

**Fig 1. Contingency tables of trial data.** The number of mosquitoes that are blood-fed ($X^f$) and the number of mosquitoes that are killed ($X^d$) are known for both data sources but, for the aggregated data we do not know directly the number of mosquitoes that are both alive and blood-fed ($X^{sf}$) (circled). We have the full complement of data from the comprehensive source.

A time-dependent logistic function is fitted to IRS impact on mosquito mortality and mosquito blood-feeding ($t$, in days).

$$X^d_{k,i} \sim Binomial(N_{k,i}, P^d_{k,i})$$

$$P^d_{k,i} = \frac{1}{1 + \exp(-(\alpha^d_k + \beta^d_k \times t_{k,i}))}$$

$$\alpha^d_k \sim Normal(\mu_{\alpha^d}, \sigma^2_{\alpha^d}), \ \beta^d_k \sim Normal(\mu_{\beta^d}, \sigma^2_{\beta^d})$$

Superscript denotes whether a mosquito will die ($d$) or survive ($s$). The subscripts denote the trial identifier $k$ and the time point at which measurements are taken $i$. We use equivalent formulae for the fed model where blood-fed ($f$) replaces mosquitoes that die ($d$). Parameters $\alpha$ and $\beta$ determine the shape of the binomial relationship and have normally distributed priors with mean $\mu$ and variance $\sigma^2$. The raw data are indicated by $X$, the count of mosquitoes that were killed (or blood-fed), and $N$, the total count of mosquitoes.

Given the absence of data on the mosquitoes that have survived and blood-fed, we can instead infer this probability by assuming independence, combine the marginal probabilities

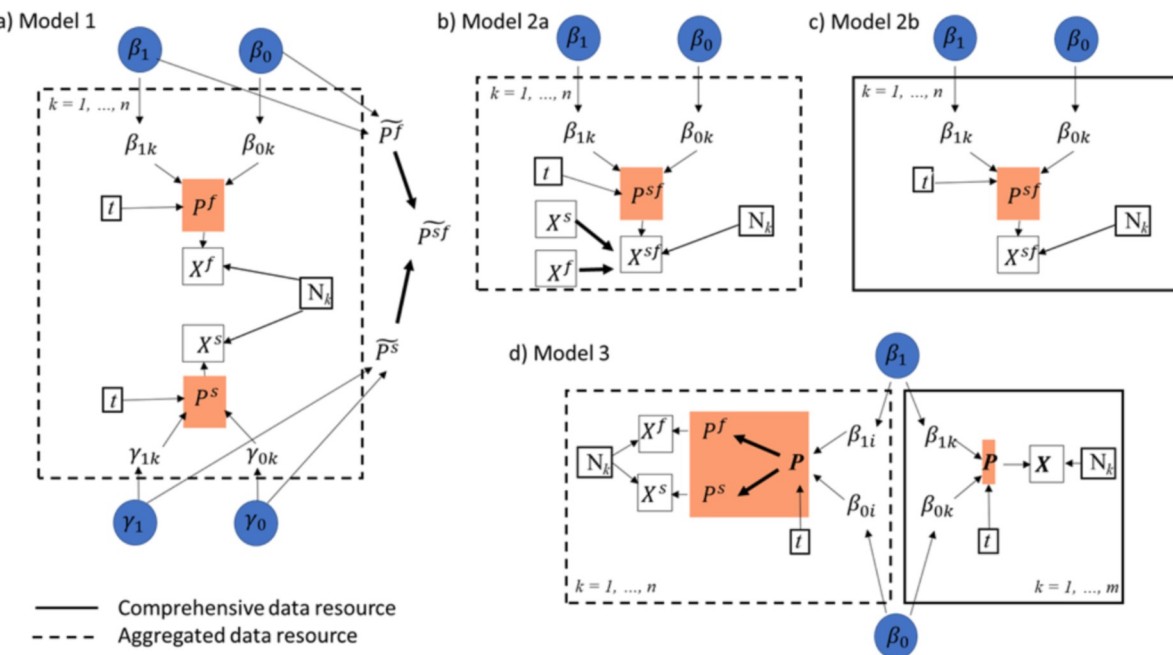

**Fig 2. Directed acyclic graphs (DAGs) for models to assess aggregated (a, b), comprehensive (c) or both aggregated and comprehensive (*d*) data.** Models 1, 2a and 2b represent models previously used in this field and Model 3 is the new approach. Model 2a depicts the original analysis in [3]. In each case, $\beta_0$ and $\beta_1$ are the shared intercept and time coefficients in the linear component of the logistic regressions, and likewise $\gamma_1$ and $\gamma_0$ for Model 1 which has separate sub-models for fed and dead. $X$ denotes the count data and $N$ are the total counts of mosquitos in each subgroup. (a) Mosquitoes are counted as killed with no information on whether they are also blood fed ($X^f$) or unfed without information that mosquitoes are also killed ($X^d$). We can infer marginal probabilities (shown in orange boxes) then (assuming independence), after estimating the probability of either outcome ($\widetilde{P^f}$ and $\widetilde{P^{sf}}$), infer the probability that mosquitoes are both alive and blood fed ($\widetilde{P^{sf}}$). b) Alternatively, we can adjust the data using the same assumption of independence prior to fitting the model and fit a logistic binomial model to the adjusted data. c) The same model structure (Model 2) can be fitted to the comprehensive data. d) Using evidence synthesis, we can learn from the comprehensive data ($N$ = 2 datasets) to infer probabilities that are supported by the aggregated data ($N$ = 23 datasets). For each dataset $k$ represents the study index and $t$ is the time point at which data are collected.

to estimate the joint probability that mosquitoes are both dead and blood-fed:

$$\widehat{P^{df}} = P^d \times P^f$$

Or those that survived and fed:

$$\widehat{P^{sf}} = P^f \times (1 - P^d)$$

In other words, we assume that mosquitoes are equally likely to have survived whether they had blood-fed or not blood-fed (Fig 3, bottom row).

**Model 2.** We contrast this with a previous approach [3], where, before fitting the time series data, the number of mosquitoes that were blood feeding ($X^f$) is adjusted by assuming that blood-feeding does not increase the probability of being killed. The blood-feeding and surviving mosquitoes are assumed to be:

$$\widehat{X^s f} = X^f \times \left(1 - \frac{X^d}{N}\right)$$

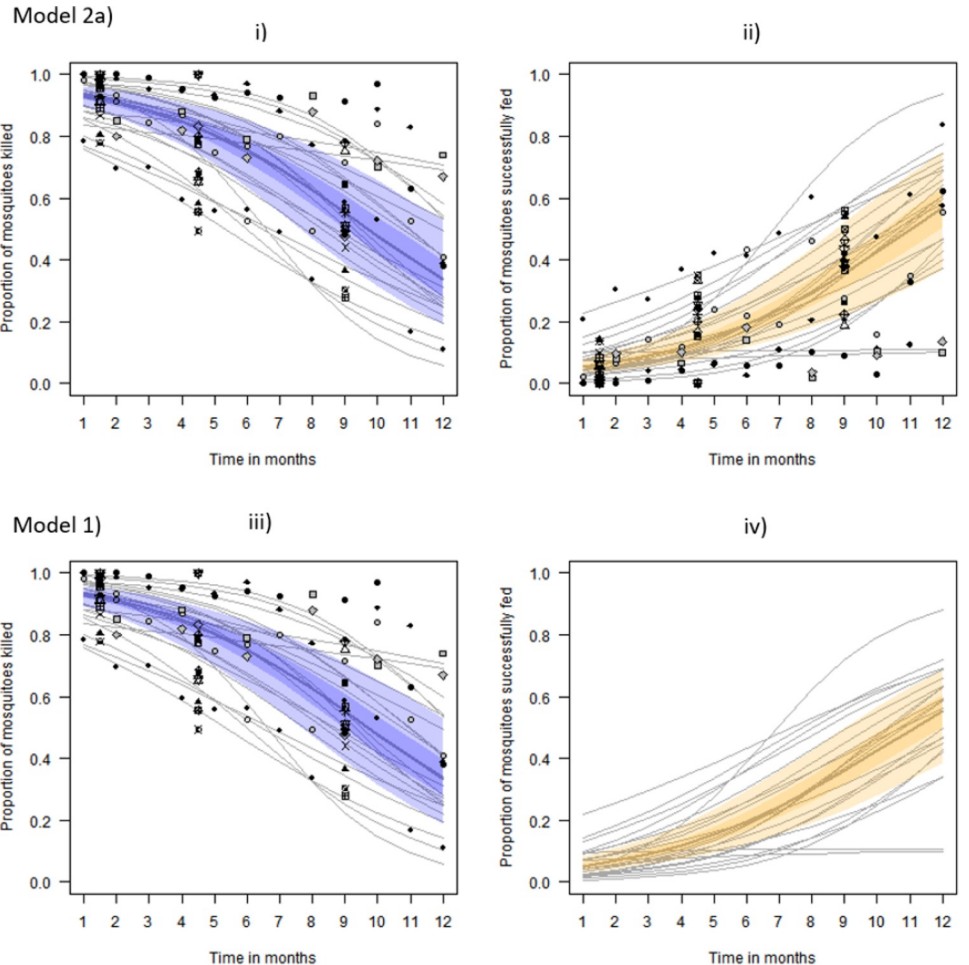

**Fig 3. Model predictions of proportion of mosquitoes killed and successfully fed.** For the best-estimate (median posterior predictive value) across all aggregate datasets, 50% (darker shaded region) and 95% (lighter shaded region) credible intervals (CrI) from the respective fits to aggregated data. Subfigures i) and ii) show results in the case of empirically disaggregating the blood fed data before fitting the model (Model 2a), and subfigures iii) and iv) show results for the case of taking this step after model fitting marginal probabilities. Data for the fits are overlaid onto the figure to demonstrate the suitability of the time-dependent functions. In each of the fits, the individual study predictions are overlaid on the figures by thin grey lines and noted by matching symbol type for each timeseries. In iv) the model was not fit to successfully fed data so has no overlaid points. *Anopheles gambiae* s.l. (circles), *An. funestus* s.l. (squares) and *An. arabiensis* (triangles) mosquitoes are noted. The data from the comprehensive source are only used in aggregated format.

A time-dependent logistic function is fitted to these adjusted data on IRS impact for mosquito survival and successful blood-feeding (*t*, in days) (Fig 3, top row):

$$X_{k,i}^{sf} \sim Binomial(N_{k,i}^{sf}, P_{k,i}^{sf})$$

$$P_{k,i}^{sf} = \frac{1}{1 + \exp(-(\alpha_k^{sf} + \beta_k^{sf} \times t_{k,i}^{sf}))}$$

$$\alpha_k^{sf} \sim Normal(\mu_{\alpha^{sf}}, \sigma_{\alpha^{sf}}^2), \ \beta_k^{sf} \sim Normal(\mu_{\beta^{sf}}, \sigma_{\beta^{sf}}^2)$$

This model is also fitted to the comprehensive data only (Fig 4, top row) to estimate the four different potential outcomes from a mosquito feeding attempt (Model 2b) shown in grey boxes in the contingency table (Fig 1).

**Model 3.** Model 3 presents the Bayesian evidence synthesis that allows inference of the parameter estimates for each potential mosquito outcome through combining both data sources (Fig 4, lower row).

AD are assumed to fit binomial distributions such that:

$$X_{k,i}^d \sim binomial(N_{k,i}, P_{k,i}^{\overline{df}} + P_{k,i}^{df})$$

$$X_{k,i}^f \sim binomial(N_{k,i}, P_{k,i}^{df} + P_{k,i}^{\overline{df}})$$

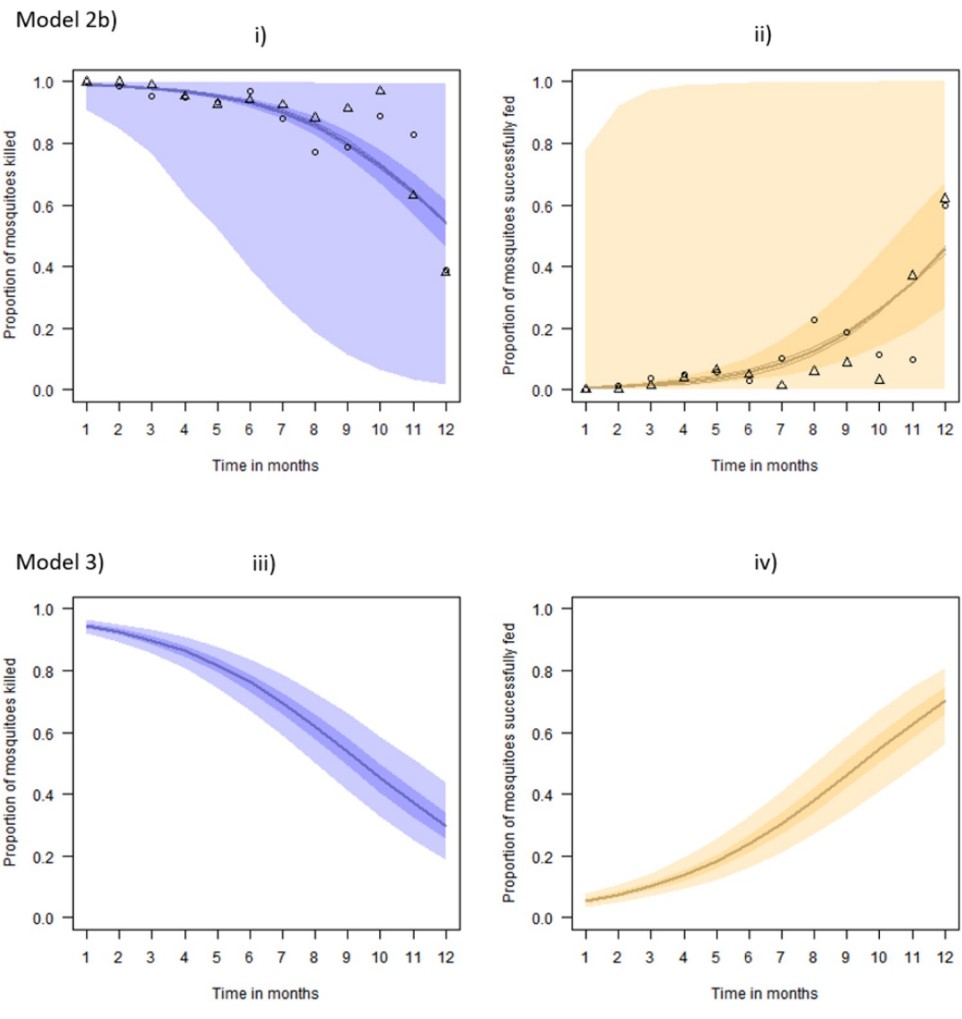

**Fig 4. Model predictions of proportion of mosquitoes killed and successfully fed.** For the best-estimate (median posterior predictive value) across all study datasets, 50% (darker shaded region) and 95% (lighter shaded region) credible intervals (CrI) from the respective fits to the comprehensive data. Subfigures i) and ii) show results using comprehensive data only (Model 2b), and subfigures iii) and iv) show results using all datasets. Data for the fits are overlaid onto subfigures i) and ii) to demonstrate the suitability of the time-dependent functions. In these fits, the individual study predictions are overlaid on the figures by thin grey lines and noted by matching symbol type for each timeseries. As Model 3 is inferred from both AD and CD sources, there are no data to overlay.

A logistic function is used to describe the time-dependent data:

$$P^1_{k,i} = \frac{1}{1 + \sum_{l=2}^{4} \exp(-(\alpha^l_k + \beta^l_k \times t^l_{k,i}))}$$

$$P^j_{k,i} = \frac{\exp(-(\alpha^j_k + \beta^j_k \times t^j_{k,i}))}{1 + \sum_{l=2}^{4} \exp(-(\alpha^l_k + \beta^l_k \times t^l_{k,i}))}, j = 2, 3, 4$$

$$\alpha^j_k \sim Normal(\mu_{\alpha j}, \sigma^2_{\alpha j}), \ \beta^j_k \sim Normal(\mu_{\beta j}, \sigma^2_{\beta j})$$

The superscript $j$ represents the possible outcomes noted in the contingency table (Fig 1). The comprehensive data are modelled similarly but using a multinomial distribution for the 4 categories and the logistic link function.

$$\boldsymbol{X} = (X^{df}, X^{sf}, X^{d\bar{f}}, X^{s\bar{f}}) \text{ and } \boldsymbol{P} = (P^{df}, P^{sf}, P^{d\bar{f}}, P^{s\bar{f}})$$

where $\boldsymbol{X} = (X^{df}, X^{sf}, X^{d\bar{f}}, X^{s\bar{f}})$ and $\boldsymbol{P} = (P^{df}, P^{sf}, P^{d\bar{f}}, P^{s\bar{f}})$.

The hyper-parameters to generate the posterior predictions for each distribution are linked across the two datasets such that they are exchangeable, i.e. for $k$ consisting of both CD and AD trials

$$\alpha^j_k \sim Normal(\mu_{\alpha j}, \sigma^2_{\alpha j}), \ \beta^j_k \sim Normal(\mu_{\beta j}, \sigma^2_{\beta j})$$

To understand better how the variability in data sources affects predictive ability of the evidence synthesis model, we contrast the predicted outcomes of the models in Fig 5. To understand how this prediction is affected by correlated data, i.e. where the predicted mosquitoes blood-feeding is correlated with the mosquito mortality outcome, we use simulated trial data from a simple copula model (Fig 6).

## Results

The model predictions of the probability that mosquitoes are killed, or alive and blood-fed, in insecticide-treated experimental huts and how these effects change over time are presented in Figs 3 and 4. The median probable outcomes are reassuringly similar however the uncertainties for different models are very different.

Where only aggregated data are available, it is only possible to predict the alive and blood-fed probability if some assumption is made about independence between feeding and dying. Assuming independence using aggregated data predicts very similar median impacts to predictions made using data with the full complement of potential outcomes from a mosquito feeding attempt. However, behavioural vector biologists report that *Anopheles gambiae* sensu lato and *An. funestus* s.l. most often rest indoors after blood-feeding [27–33], which indicates that blood-fed mosquitoes are possibly more likely to be exposed to insecticides sprayed onto indoor surfaces.

Only 2 datasets with a full breakdown of potential outcomes were available (Table 1). The study generating these data was conducted in Benin, and tested *An gambiae* s.l. mosquitoes in West African huts with both walls and ceilings sprayed. The Benin study provided no netting for sleepers. Almost all mosquitoes blood-feed throughout the studies (median estimate across studies and months since spraying; 91.5%, 95%CI; 68.4% - 100% mosquitoes are blood-fed).

We can see in Fig 5 that the difference between the different predictions can be significant over longer periods of time. The model results using comprehensive data only are very

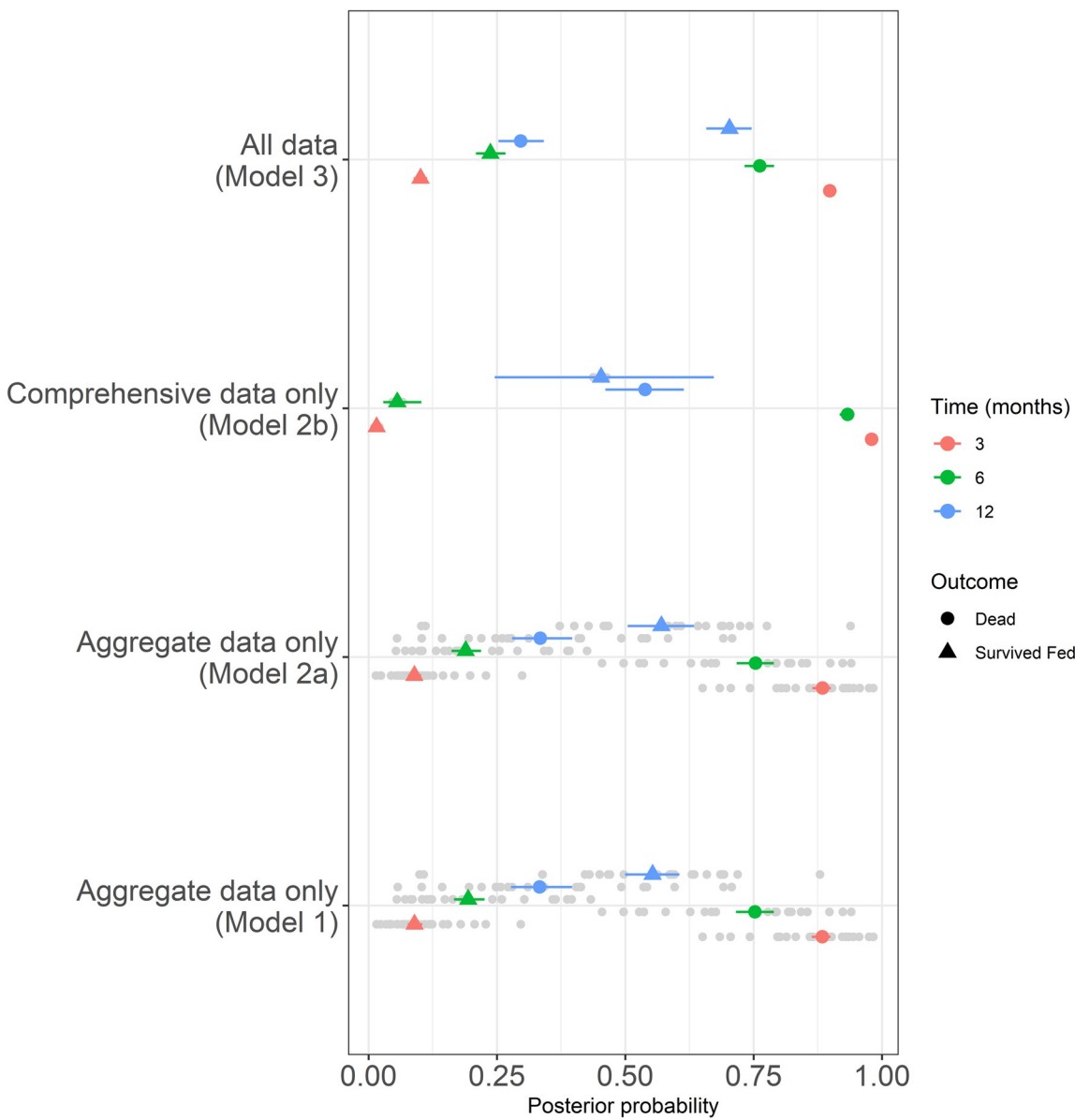

**Fig 5. A forest plot of posterior predictions for all models considered.** Showing median, 25% and 75% percentiles for dead and survived fed probabilities across all study datasets at 3, 6 and 12 months. Individual study predictions are shown with grey points.

uncertain, largely due to few time series available. However, these data provide worthwhile additional information when combined with the aggregate data such that the proportion survived and fed at 12 months could be approximately 10% higher than the aggregate data only model suggests.

A major advantage of the Bayesian evidence synthesis over the alternative methods is seen by the tightened uncertainty bounds predicted (see Fig 5). This is driven by the additional data available for this statistical framework. To understand how this prediction is affected by correlated data sources we explore the correlation coefficient using simulated trial data from a simple copula model. We can see from Fig 6a-6c that when mortality and blood-feeding are positively correlated, the number of successfully fed mosquitoes is under-estimated. This is our data situation. Alternatively, when mortality and blood-feeding are negatively correlated

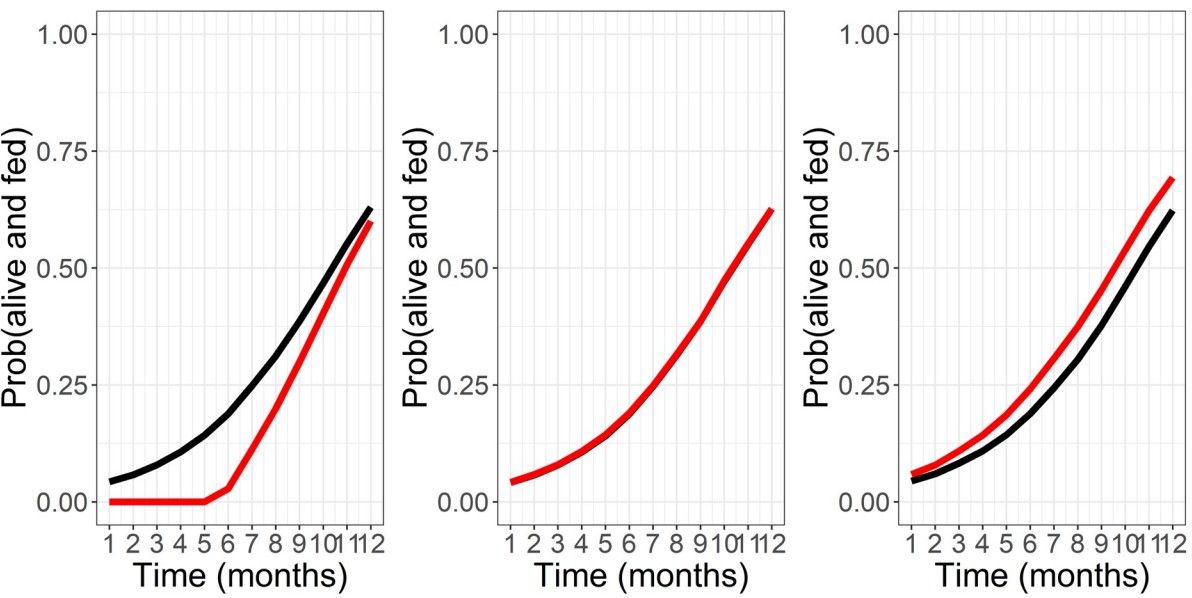

**Fig 6. Probability of success (alive) and fed over time for independent and joint models using simulated trial data.** A copula method is used to impose correlation. The black lines are the Independent estimates and the red lines are the correct joint estimates. The estimates are based on the empirical marginal probabilities. a) negative correlation. b) independent c) positive correlation (as is the case for the experimental hut data examined here).

the number of successfully fed is over-estimated. Increasing data in the comprehensive resource will always be beneficial to the accuracy of the predictions.

## Discussion

Concerns around mosquito resistance to pyrethroid insecticides has accelerated development of alternative insecticide products to help mitigate against any diminishing impact from pyrethroid-based vector control. By necessity, these products have different mechanisms of action so that rigorous testing is required prior to determining their potential benefit. This generates a wealth of data. Alongside this, statistical methods are developing and their availability to biological scientists is increasing through software languages such as R, BUGS and Stan. We demonstrate with these data how statistical models can be adapted and developed to maximise performance of inference from different but related data sources. This is all the more important since there are ethical decisions to be made when conducting such trials that result in differences: a key example is the debate about providing an untreated (sometimes holed) net for the study participant present to lure mosquitoes indoors during an IRS hut study. This is necessary since providing an intact net during an IRS study will prevent most blood-feeding which will then affect mosquito behaviour as IRS is designed primarily to target mosquitoes that are resting on walls after blood-feeding. In an IRS hut without a net most mosquitoes will feed and then rest on walls to digest the blood meal and are then killed by the IRS. A further complication is that, if a mosquito encounters a net, she may alight on the wall unfed and be killed by the IRS in which case the proportion of mosquitoes that are unfed-dead will be higher, as will be the proportion unfed-live mosquitoes, which may go on to feed elsewhere and transmit malaria.

Bayesian evidence synthesis (Model 3) allows both the most available, aggregated data and most accurate, comprehensive data to inform predictions of probable outcomes of mosquito feeding attempts. We show that this approach can refine predictions and tighten uncertainty.

In practical terms, this may significantly increase or decrease the predicted proportions and so potentially impact decision making. The volume of aggregated data relative to comprehensive data means that the inference is shrunk towards that of Models 1 and 2a (Fig 3) which use aggregated data alone. With a greater volume of comprehensive data, which would be the ideal, these predictions might be altered. Simulating a correlated Bernoulli random variable allows us to understand how these biases play out; essentially, where probabilities are positively correlated (in our case, the probability that mosquitoes will be killed is correlated with the probability that mosquitoes will also be fed), we would expect the evidence synthesis to underestimate the probability of mosquitoes blood-feeding and surviving. Conversely, were these probabilities negatively correlated then those mosquitoes surviving having blood-fed would be overestimated. In the context of IRS impact on mosquito behaviour, both scenarios are problematic. The first would overestimate the impact of the intervention while the second would undersell its potential. Acknowledging these effects of the statistical approach are therefore crucial.

We recognise that other, related modelling choices were available. For example, in terms of data, we could have aggregated the comprehensive data and combined these with the originally aggregated data to create a new aggregated data set. In terms of modelling, the aggregate data-only analysis assumed independent Binomial models for death and successfully fed. We could have estimated the two outcomes jointly as in the full Bayesian evidence synthesis model. However, the aim of this work was to compare approaches performed previously [16,17,26] against the full Bayesian evidence synthesis approach, which we consider the best option given all data and can be used for understanding impact.

Alternative approaches to evidence synthesis are published which could be explored for their suitability to the peculiarities of the example used here, such as partial reconstruction of comprehensive data [34]. Another alternative is to include an indicator to the linear predictor to note those studies with additional data or potentially those where some assumption like independence is made *a priori* and allow them to have more weight in the ultimate predictions [35]. There are also sources of uncertainty in the data that we do not explore here including differences in wall surface, the provision of, and number of holes in, untreated mosquito nets for volunteer sleepers, the geographic location of the hut trial and respective mosquito species represented. Other studies have begun to consider the scale and implications of these differences [36–38].

The approach detailed here can be thought of generally as a method of model calibration using indirect data [39]. In our case, this is using marginal (aggregate) and joint (comprehensive) data to inform a joint distribution, but the problem can be more general than this and the data less directly relevant. The approach provides a principled framework to synthesise multiple, different data sets. To date, Bayesian evidence synthesis has been applied to a range of problems, for example, to estimating HIV and Hepatitis C prevalence [40,41].

Given the increased capacity to collate multiple data and the increased access to statistical software, it is increasingly important to explore how different methods compare and infer biologically relevant conclusions. Bayesian evidence synthesis provides a robust statistical approach when different resources are available but may be biased toward the dataset with the greatest absolute quantity of information.

## Supporting information

**S1 Appendix. Datasets for analyses.**
(XLSX)

**S2 Appendix. Modelling details, including BUGS and Stan code.**
(PDF)

## Acknowledgments

We thank all data teams, those volunteering in the experimental huts, to those coordinating the trials.

## Author Contributions

**Conceptualization:** Nathan Green, Boulais Yovogan, Ellie Sherrard-Smith.

**Data curation:** Fiacre Agossa, Richard Oxborough, Jovin Kitau, Edi Constant, Mark Rowland, Emile F. S. Tchacaya, Koudou G. Benjamin, Thomas S. Churcher.

**Formal analysis:** Nathan Green, Ellie Sherrard-Smith.

**Investigation:** Nathan Green, Ellie Sherrard-Smith.

**Methodology:** Nathan Green, Michael Betancourt, Ellie Sherrard-Smith.

**Software:** Nathan Green, Michael Betancourt, Ellie Sherrard-Smith.

**Validation:** Nathan Green, Ellie Sherrard-Smith.

**Visualization:** Nathan Green, Ellie Sherrard-Smith.

**Writing – original draft:** Nathan Green, Fiacre Agossa, Boulais Yovogan, Richard Oxborough, Jovin Kitau, Pie Müller, Edi Constant, Mark Rowland, Emile F. S. Tchacaya, Koudou G. Benjamin, Thomas S. Churcher, Michael Betancourt, Ellie Sherrard-Smith.

**Writing – review & editing:** Nathan Green, Fiacre Agossa, Boulais Yovogan, Richard Oxborough, Jovin Kitau, Edi Constant, Mark Rowland, Emile F. S. Tchacaya, Koudou G. Benjamin, Thomas S. Churcher, Michael Betancourt, Ellie Sherrard-Smith.

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
