## [Decision Letter · Decision Letter 0]

10 Nov 2021

PONE-D-21-15152An evidence synthesis approach for combining different data sources illustrated using entomological efficacy of insecticides for indoor residual sprayingPLOS ONE

Dear Dr. Green,

Thank you for submitting your manuscript to PLOS ONE. After careful consideration, we feel that it has merit but does not fully meet PLOS ONE’s publication criteria as it currently stands. Therefore, we invite you to submit a revised version of the manuscript that addresses the points raised during the review process.

We look forward to receiving your revised manuscript.

Kind regards,

Monika Gulia-Nuss, PhD

Academic Editor

PLOS ONE

Journal Requirements:

2. In the Methods section, please provide additional details regarding the methodology used to identify the three datasets included in the analysis.

3. As explained in PLOS ONE's manuscript guidelines (http://journals.plos.org/plosone/s/submission-guidelines#loc-references), we do not allow citation of, or reliance on, unpublished work. Please provide the relevant information in the manuscript and/or a Supporting Information file; if the manuscript described in line 118 is accepted for publication while your PLOS ONE submission is under review, you can remove the Supporting Information file and revert to the citation in your Methods section before acceptance/publication of the current submission.

4. Please update your submission to use the PLOS LaTeX template. The template and more information on our requirements for LaTeX submissions can be found at http://journals.plos.org/plosone/s/latex.

5. Please note that in order to use the direct billing option the corresponding author must be affiliated with the chosen institute. Please either amend your manuscript to change the affiliation or corresponding author, or email us at plosone@plos.org with a request to remove this option.

7. Please amend the manuscript submission data (via Edit Submission) to include authors Fiacre Agossa, Boulais Yovogon, Richard Oxborough, Jovin Kitau, Pie Müller,Edi Constant, Mark Rowland, Emile FS Tchacaya, Koudou G Benjamin, Thomas S Churcher, Michael Betancourt, Ellie Sherrard-Smith.

8. Please upload a copy of Supporting Material 1 ,2, and Supplementary data file 1 which you refer to in your text on page 20.

Reviewers' comments:

Reviewer's Responses to Questions

**Comments to the Author**

1. Is the manuscript technically sound, and do the data support the conclusions?

Reviewer #1: Partly

Reviewer #2: Yes

2. Has the statistical analysis been performed appropriately and rigorously? 

Reviewer #1: I Don't Know

Reviewer #2: Yes

3. Have the authors made all data underlying the findings in their manuscript fully available?

Reviewer #1: Yes

Reviewer #2: Yes

4. Is the manuscript presented in an intelligible fashion and written in standard English?

Reviewer #1: Yes

Reviewer #2: Yes

5. Review Comments to the Author

Reviewer #1: The authors present a well written manuscript on use of Bayesian evidence synthesis for predicting effects of indoor residual spraying. The method they propose allows combination of different data types. However, this is also the pitfall of the manuscript. As their Bayesian evidence synthesis model uses different data than the models to which is it is compared, it is challenging to draw conclusions towards differences in performance.

Line 38, this reviewer does not see a need for using the registered trademark symbol in the mention of the brand name of the product under study in the abstract. Instead, the authors may use a description of the product using the trivial name, e.g. 300g/L Pirimiphos-methyl

Lines 43-44 “The evidence synthesis model was most robust at predicting the probability of mosquitoes dying or surviving and blood-feeding.“ Most robust compared to what?

Results

Line 260: Reference to figure 5 where figure 6 is meant

Figures 3& 4: at the resolution presented, the distinction between dotted and dashed lines mentioned in the legend for data from Benin and Cote d’Ivoire, respectively, is not clear

Figure 5: the legend indicates that the plot shows predictions for all models but the annotations on the y-axis are not clear. What is meant with ‘aggregate after’ and aggregate before’? Where can we see the predictions for models 1, 2 and 3?

Line 252-254. Indeed the uncertainty is lower for the outcomes predicted for ‘all’ compared to the other groups. But is this an intrinsic property of the model or just a result of the fact that more data is fed into this model?

It is not immediate evident from the manuscript that model 3 is superior over the other models. The predictions presented in Figures 3A&C are not very different from those presented in Figure 4C, and uncertainties appear similar.

The claim that the Bayesian evidence synthesis provides the best modelling option is not fully substantiated. The manuscript lacks a ‘ground truth’ to which data is compared. The approach described by the authors in lines 302-308, to created an aggregated dataset from comprehensive data, would provide a means to generate such a ground truth. It is advised that the authors include this analysis in a new version of the manuscript

Conflict of interest statement

Please clarify whether the authors affiliated to private entities (e.g. Abt Associates and Symplectromorphic) hold stock or performed commercial services

Reviewer #2: Review of the article: “An evidence synthesis approach for combining different data sources illustrated using entomological efficacy of insecticides for indoor residual spraying”

The authors describe a Bayesian evidence synthesis model and framework, building on their previously described models, to combine aggregated and comprehensive data sources for outcomes relating to mosquito feeding. The manuscript outlines the benefits and limitations of previous models and the new modelling framework, through the analysis of 23 aggregated datasets and 3 comprehensive datasets.

The evidence synthesis approach is clearly described via DAGs and within the text, and the statistical differences between the previous models and the new approach are generally clearly outlined.

My comments mainly relate to the presentation of some of the results:

1) One comment for the Editor, I was unable to access the Supplementary Material and Supplementary Data Files in the Editorial Manager system so my review does not include these materials

2) Throughout most of the manuscript, including the methods and results sections, the authors refer to ‘aggregated’ and ‘comprehensive’ data (i.e. referring to whether the outcomes related to mosquito feeding are published in a combined format or separated by each possible feeding outcome).

In the Introduction, rather than ‘comprehensive’ data, the authors refer to ‘individual-level data’ (presumably referring to outcomes reported at an individual level, rather than mosquito level data, i.e. individual participant data, being available).

I suggest sticking to the term ‘comprehensive data’ throughout and defining this term in the Introduction, to avoid confusion with individual participant data synthesis, such as IPD meta-analysis, which is a different framework entirely.

3) Some of the labelling of Figures in the text seems to be incorrect:

a. Line 139: should this read Figure 2a, Model 1?

b. Line 155: Which Model 2 is referred to here, Model 2a?

c. Line 218: I'm not following how Figure 5 shows this, should this be Figure 6?

d. Line 260: Should this read Figures 6a-c?

4) The assumption have needed to be made in previous models (e.g. Model 1), that mosquitos are equally likely to have been killed and blood-fed is described as a simplifying assumption within the Introduction. While it may be a simple assumption, I guess the key question is whether it is a reasonable assumption to make clinically or not.

If I understand correctly, part of the rationale for the new Bayesian evidence synthesis framework is that this assumption is not required? But if this ‘simple’ assumption is actually reasonable, then is the added complexity (presumably) of this new framework a reasonable trade-off to not make the assumption?

I think what I’m asking here is for a little more clarity on what the benefits are (if any) of the new modelling framework compared to the previous models with respect to this assumption?

5) I found the legends of most of the Figures quite unclear, specifically what each individual Figure (i.e. a – d) was showing me, i.e. which model (1, 2a/b, 3), aggregated data, comprehensive data or both etc. Although some of this information can be deduced from the text, it would be helpful to have fully descriptive legends. Specifically:

a. Figure 2: A clear statement of which of these are the previously described models and which of these is the new model would be helpful. Also, not all notation is defined – e.g. the betas, Y's and N's. Again while some of this can be deduced, a full list would be helpful for completeness and particularly for readers to who may not be experts in DAGs

b. Figure 3 and 4: While the general summaries are quite clear, i.e. what the shaded regions shown etc. it isn’t clear what a-d individually show (model (1, 2a/b, 3), aggregated data, comprehensive data or both etc.). I am also unable to see dotted lines and dashed lines corresponding to Benin and Côte d'Ivoire respectively on any of these Figures.

c. Figure 5: Not sure I follow what aggregate data before and aggregate data after means

6) Line 248-251: “The model using comprehensive data only is very uncertain. However, these

data provide worthwhile additional information when combined with the aggregate data such that the proportion survived and fed at 12 months could be approximately 10% higher than the aggregate data only model..”

This is eluded to in the text below but this uncertainty is likely due to the small amount comprehensive data and the large amount of aggregate data in comparison, rather than directly due to the models themselves? Perhaps it would be better to describe that the results or estimates using comprehensive data only are uncertain, rather than the models themselves?

7) A general query, related somewhat to the discussion. The series of models here are clearly suitable for addressing the particular question in hand related to mosquito feeding but are these models ‘tailor made’ only for this question? Are there any features of the models which could be generalizable to a wider set of research questions or contexts? Bringing this out a little more in the discussion may be helpful

6. PLOS authors have the option to publish the peer review history of their article (what does this mean?). If published, this will include your full peer review and any attached files.

Reviewer #1: No

Reviewer #2: **Yes: **Sarah J Nevitt

---

## [Author Response · Author response to Decision Letter 0]

11 Jan 2022

We thank both reviewers for insightful critique of this work which we think has greatly improved it.

Reviewer #1:

The authors present a well written manuscript on use of Bayesian evidence synthesis for predicting effects of indoor residual spraying. The method they propose allows combination of different data types. However, this is also the pitfall of the manuscript. As their Bayesian evidence synthesis model uses different data than the models to which is it is compared, it is challenging to draw conclusions towards differences in performance.

Thank you, however, we disagree here that this is a pitfall because the point of the paper is to explore how the evidence synthesis model is able to incorporate additional comprehensive data to augment the inference, which by definition, we cannot use within the Model 1 framework. We have endeavoured to make the approaches clearer. As this is a statistical analysis, all models will benefit from more relevant data and particularly if these data contain information about additional structure. This work enables us to use all available data to make the most reliable inference. Previously, researchers in this field had to either aggregate data where comprehensive data were sparse, or disregard some resources altogether. We have tried to make this clearer and have added to the introduction:

Lines 101-105: “Our recent assessment of IRS products made the assumption that mosquitoes were equally likely to have blood-fed and survived or blood-fed and died on entering a sprayed hut(3). However, IRS exploits the resting behaviour of mosquitoes after feeding, so we need a method to capture the likely higher mortality in fed mosquitoes.”

To the method:

Lines 121-122: “We apply the models to an empirical dataset to explore how mosquito outcomes can be interpreted from experimental hut trials testing the efficacy of an IRS product.”

And:

Lines 235-243: “To understand better how the variability in data sources affects predictive ability of the evidence synthesis model, we contrast the predicted outcomes of the models in Figure 5. To understand how this prediction is affected by correlated data sources, i.e. where the predicted mosquitoes blood-feeding is correlated with the mosquito mortality outcome, we explore the correlation coefficient using simulated trial data from a simple copula model (Figure 6).”

Line 38, this reviewer does not see a need for using the registered trademark symbol in the mention of the brand name of the product under study in the abstract. Instead, the authors may use a description of the product using the trivial name, e.g. 300g/L Pirimiphos-methyl.

We are happy to remove and have followed the advice from the reviewer.

Lines 43-44 “The evidence synthesis model was most robust at predicting the probability of mosquitoes dying or surviving and blood-feeding.“ Most robust compared to what?

We agreed this was unclear and changed to 'has the smallest uncertainty'.

Results

Line 260: Reference to figure 5 where figure 6 is meant

Thanks for noticing this. Now corrected.

Figures 3& 4: at the resolution presented, the distinction between dotted and dashed lines mentioned in the legend for data from Benin and Cote d’Ivoire, respectively, is not clear

We agree this was not clear and so we have removed reference to the study country as this is not important for the analysis. We have re-labelled the Figures for clarity to make it clear which model is represented in each row for Figures 3 and 4.

We have re-written the legends for clarity as follows:

“Fig 3: Model predictions of proportion of mosquitos killed and successfully fed for the best-estimate (median posterior predictive value) across all aggregate datasets, 50% (darker shaded region) and 95% (lighter shaded region) credible intervals (CrI) from the respective fits to aggregated data. Subfigures i) and ii) show results in the case of empirically disaggregating the blood fed data before fitting the model (Model 2a), and subfigures iii) and iv) show results for the case of taking this step after model fitting marginal probabilities. Data for the fits are overlaid onto the figure to demonstrate the suitability of the time-dependent functions. In each of the fits, the individual study predictions are overlaid on the figures by thin grey lines and noted by matching symbol type for each timeseries. In iv) the model was not fit to successfully fed data so has no overlaid points. Anopheles gambiae s.l. (circles), An. funestus s.l. (squares) and An. arabiensis (triangles) mosquitoes are noted. The data from the comprehensive source are only used in aggregated format.”

“Fig 4: Model predictions of proportion of mosquitos killed and successfully fed for the best-estimate (median posterior predictive value) across all study datasets, 50% (darker shaded region) and 95% (lighter shaded region) credible intervals (CrI) from the respective fits to the comprehensive data. Subfigures i) and ii) show results using comprehensive data only (Model 2b), and subfigures iii) and iv) show results using all datasets. . Data for the fits are overlaid onto subfigures i) and ii) to demonstrate the suitability of the time-dependent functions. In these fits, the individual study predictions are overlaid on the figures by thin grey lines and noted by matching symbol type for each timeseries. As Model 3 is inferred from both AD and CD sources, there are no data to overlay.”

Figure 5: the legend indicates that the plot shows predictions for all models but the annotations on the y-axis are not clear. What is meant with ‘aggregate after’ and aggregate before’? Where can we see the predictions for models 1, 2 and 3?

We agree that this was unclear. In this instance, we made the assumption that the same proportion of mosquitoes would have been killed had they blood fed, as would have survived having blood fed. To fit Model 2a using this assumption, we manipulate the data to provide a ‘count’ of mosquitoes that have fed and survived (as the numbers fed and surviving plus dead should sum to the total or less than this count). In Model 1, we obtain this using probabilities after fitting the statistical model.

To clarify, we have updated the y-axis as suggested. Now we make explicit reference to the data used and the model employed for each fit.

“Fig 5: A forest plot of posterior predictions for all models considered, showing median, 25% and 75% percentiles for dead and survived fed probabilities across all study datasets at 3, 6 and 12 months. Individual study predictions are shown with grey points.”

Line 252-254. Indeed the uncertainty is lower for the outcomes predicted for ‘all’ compared to the other groups. But is this an intrinsic property of the model or just a result of the fact that more data is fed into this model?

We agree and have added (now to Line 272-274): “The model results using comprehensive data only are very uncertain, largely due to few time series available”

It is not immediate evident from the manuscript that model 3 is superior over the other models. The predictions presented in Figures 3A&C are not very different from those presented in Figure 4C, and uncertainties appear similar. The claim that the Bayesian evidence synthesis provides the best modelling option is not fully substantiated. The manuscript lacks a ‘ground truth’ to which data is compared. The approach described by the authors in lines 302-308, to created an aggregated dataset from comprehensive data, would provide a means to generate such a ground truth. It is advised that the authors include this analysis in a new version of the manuscript.

In our case, we consider it is better to use a model that principally and appropriately incorporates all the different data sources. Since the results are different for the different models then that is an argument to use the evidence synthesis one. It shows that more information (the inclusion of both AD and CD sources) is modifying the outcomes predicted. 

The aggregate models assume independence which is unlikely. The question is are these data (the proportion of mosquitoes feeding, and those that are dying) correlated and can we use the information about this in the comprehensive data to help out?

From the covariance formula, we get: E(XY) = E(X)E(Y) + cov(X,Y)

This gives, p(dead and fed) = p(dead)p(fed) + cov(I_dead, I_fed)

And so we don’t need to simulate a trial to show this. The aggregate models do not properly represent the correlation, but the evidence synthesis model has a latent dead and fed category. This also then means that the aggregate and comprehensive data can be fit in the same model.

We certainly appreciate the suggestion from the reviewer and so we now show the effect for differently correlated data by employing a copula approach to simulate the trial data. This enables us to simulate uncorrelated and highly correlated dead and fed trial data. This gives the following figure, which we include as Figure 6. The red curves are the correct joint distribution p(fed, dead) and the black curves are the empirical p(fed)p(dead) estimates, which we would have obtained with the naïve model.

”Fig 6. Probability of success (alive) and fed over time for independent and joint models using simulated trial data. A copula method is used to impose correlation. The black lines are the Independent estimates and the red lines are the correct joint estimates. The estimates are based on the empirical marginal probabilities. a) negative correlation. b) independent c) positive correlation (as is the case for the experimental hut data examined here).”

We have explained this in more detail. What we see is how the predictions may be affected given the correlation between data outcomes. That is, if blood feeding mosquitoes are more likely to die – if data outcomes are positively correlated – then we can show this would under-estimate the number of successfully fed mosquitoes (this is likely the case for our data). As more comprehensive data becomes available, we can now use the evidence synthesis model to learn this association more robustly, with a better inference of the impacts of the IRS (and other interventions where the similar challenge exists) moving forward. 

Lines 277-289: “A major advantage of the Bayesian evidence synthesis over the alternative methods is seen by the tightened uncertainty bounds predicted (see Figure 5). This is driven by the additional data available for this statistical framework. To understand how this prediction is affected by correlated data sources we explore the correlation using simulated trial data from a simple copula model. We can see from Figure 6a-c that when mortality and blood-feeding are positively correlated, the number of successfully fed mosquitoes is under-estimated. This is our data situation. Alternatively, when mortality and blood-feeding are negatively correlated the number of successfully fed is over-estimated. Increasing data in the comprehensive resource will always be beneficial to the accuracy of the predictions.”

-----

Reviewer #2: Review of the article: “An evidence synthesis approach for combining different data sources illustrated using entomological efficacy of insecticides for indoor residual spraying”

The authors describe a Bayesian evidence synthesis model and framework, building on their previously described models, to combine aggregated and comprehensive data sources for outcomes relating to mosquito feeding.

The manuscript outlines the benefits and limitations of previous models and the new modelling framework, through the analysis of 23 aggregated datasets and 3 comprehensive datasets.

The evidence synthesis approach is clearly described via DAGs and within the text, and the statistical differences between the previous models and the new approach are generally clearly outlined.

My comments mainly relate to the presentation of some of the results:

1) One comment for the Editor, I was unable to access the Supplementary Material and Supplementary Data Files in the Editorial Manager system so my review does not include these materials.

Apologies, the Supplementary Data (S1 Appendix) containing both complete data sources, and Supplementary Material (S2 Appendix) which contains the code, are now included.

2) Throughout most of the manuscript, including the methods and results sections,

the authors refer to ‘aggregated’ and ‘comprehensive’ data

(i.e. referring to whether the outcomes related to mosquito feeding are published in a combined format or separated by each possible feeding outcome). In the Introduction, rather than ‘comprehensive’ data, the authors refer to ‘individual-level data’ (presumably referring to outcomes reported at an individual level, rather than mosquito level data, i.e. individual participant data, being available). I suggest sticking to the term ‘comprehensive data’ throughout and defining this term in the Introduction, to avoid confusion with individual participant data synthesis, such as IPD meta-analysis, which is a different framework entirely.

Thank you. This is a good point. By comprehensive data in our case we don’t actually require individual level data although this would be the ideal. We have changed all mention of ILD and individual-level as suggested.

3) Some of the labelling of Figures in the text seems to be incorrect:

a. Line 139: should this read Figure 2a, Model 1?

b. Line 155: Which Model 2 is referred to here, Model 2a?

c. Line 218: I'm not following how Figure 5 shows this, should this be Figure 6?

d. Line 260: Should this read Figures 6a-c?

Thank you for picking this up. Now corrected. We have further adjusted the Figures to make it clearer as to which Model and which data are being considered in each case. We have updated the legends accordingly. 

4) The assumption have needed to be made in previous models (e.g. Model 1), that mosquitos are equally likely to have been killed and blood-fed is described as a simplifying assumption within the Introduction. While it may be a simple assumption, I guess the key question is whether it is a reasonable assumption to make clinically or not. If I understand correctly, part of the rationale for the new Bayesian evidence synthesis framework is that this assumption is not required? But if this ‘simple’ assumption is actually reasonable, then is the added complexity (presumably) of this new framework a reasonable trade-off to not make the assumption?

I think what I’m asking here is for a little more clarity on what the benefits are (if any) of the new modelling framework compared to the previous models with respect to this assumption?

We agree this is an important clarification. IRS is designed to work by targeting the mosquito pathway of activity observed where having fed, mosquitoes then rest on a surface. So, it is clinically likely that the assumption of feeding and dying being independent misrepresents what might be seen in reality. If mosquitoes rest after feeding it may be reasonable to assume more deaths post feeding, and therefore the outcomes to be positively correlated. This is not possible to see with the data when aggregated. So, the benefits are to have an approach that can learn from all the data available, until comprehensive data are numerous enough to use to explore this unknown more thoroughly.

We have added to the introduction:

Lines 101-105: “Our recent assessment of IRS products made the assumption that mosquitoes were equally likely to have blood-fed and survived or blood-fed and died on entering a sprayed hut(3). However, IRS exploits the resting behaviour of mosquitoes after feeding, so we need a method to capture the likely higher mortality in fed mosquitoes.”

To the method:

Lines 121-122: “We apply the models to an empirical dataset to explore how mosquito outcomes can be interpreted from experimental hut trials testing the efficacy of an IRS product.”

And:

Lines 235-243: “To understand better how the variability in data sources affects predictive ability of the evidence synthesis model, we contrast the predicted outcomes of the models in Figure 5. To understand how this prediction is affected by correlated data sources, i.e. where the predicted mosquitoes blood-feeding is correlated with the mosquito mortality outcome, we explore the correlation coefficient using simulated trial data from a simple copula model (Figure 6).”

This is further complicated depending on the way that the experiment was conducted which is beyond the scope of our contribution (particularly given we have 2 time series from the same study comprising our comprehensive data). In these experimental hut studies, volunteers are provided with either no net, an untreated and holed net, or an untreated net without holes. This clearly will interact with the observed feeding outcomes. With more data, this is something we could eventually look at with the evidence synthesis framework. We discuss these issues briefly.

Lines 300-310: “This is all the more important since there are ethical decisions to be made when conducting such trials that result in differences: a key example is the debate about providing an untreated (sometimes holed) net for the study participant present to lure mosquitoes indoors during an IRS hut study. This is necessary since providing an intact net during an IRS study will prevent most blood-feeding which will then affect mosquito behaviour as IRS is designed primarily to target mosquitoes that are resting on walls after blood-feeding. In an IRS hut without a net most mosquitoes will feed and then rest on walls to digest the blood meal and are then killed by the IRS. A further complication is that, if a mosquito encounters a net, she may alight on the wall unfed and be killed by the IRS in which case the proportion of mosquitoes that are unfed-dead will be higher, as will be the proportion unfed-live mosquitoes, which may go on to feed elsewhere and transmit malaria.”

5) I found the legends of most of the Figures quite unclear, specifically what each individual Figure (i.e. a – d) was showing me, i.e. which model (1, 2a/b, 3), aggregated data, comprehensive data or both etc. Although some of this information can be deduced from the text, it would be helpful to have fully descriptive legends. Specifically:

a. Figure 2: A clear statement of which of these are the previously described models and which of these is the new model would be helpful. Also, not all notation is defined – e.g. the betas, Y's and N's. Again while some of this can be deduced, a full list would be helpful for completeness and particularly for readers to who may not be experts in DAGs

b. Figure 3 and 4: While the general summaries are quite clear, i.e. what the shaded regions shown etc. it isn’t clear what a-d individually show (model (1, 2a/b, 3), aggregated data, comprehensive data or both etc.). I am also unable to see dotted lines and dashed lines corresponding to Benin and Côte d'Ivoire respectively on any of these Figures.

c. Figure 5: Not sure I follow what aggregate data before and aggregate data after means

We agree this was unclear and thank the reviewer(s) for highlighting this. We have:

 - Changed the labelling on the figures to make it clear which model and data source we are using.

 - Ensured the notation used in the DAGs, Table and methods aligns.

 - Rewritten the legends (please see below).

 - Removed reference to study country because this is not directly relevant to this piece of work.

“Fig 2: Directed acyclic graphs (DAGs) for models to assess aggregated (a, b), comprehensive (c) or both aggregated and comprehensive (d) data. Models 1, 2a and 2b represent models previously used in this field and Model 3 is the new approach. Model 2a depicts the original analysis in (3). In each case, β_0 and β_1 are the shared intercept and time coefficients in the linear component of the logistic regressions, and likewise γ_1 and γ_0 for Model 1 which has separate sub-models for fed and dead. X denotes the count data and N are the total counts of mosquitos in each subgroup. (a) Mosquitoes are counted as killed with no information on whether they are also blood fed (X^f) or unfed without information that mosquitoes are also killed (X^d). We can infer marginal probabilities (shown in orange boxes) then (assuming independence), after estimating the probability of either outcome ( (P^f ) ~ and (P^sf ) ~), infer the probability that mosquitoes are both alive and blood fed ((P^sf ) ~). b) Alternatively, we can adjust the data using the same assumption of independence prior to fitting the model and fit a logistic binomial model to the adjusted data. c) The same model structure (Model 2) can be fitted to the comprehensive data. d) Using evidence synthesis, we can learn from the comprehensive data (N = 2 datasets) to infer probabilities that are supported by the aggregated data (N = 23 datasets). For each dataset k represents the study index and t is the time point at which data are collected.”

Fig 3: Model predictions of proportion of mosquitos killed and successfully fed for the best-estimate (median posterior predictive value) across all aggregate datasets, 50% (darker shaded region) and 95% (lighter shaded region) credible intervals (CrI) from the respective fits to aggregated data. Subfigures i) and ii) show results in the case of empirically disaggregating the blood fed data before fitting the model (Model 2a), and subfigures iii) and iv) show results for the case of taking this step after model fitting marginal probabilities. Data for the fits are overlaid onto the figure to demonstrate the suitability of the time-dependent functions. In each of the fits, the individual study predictions are overlaid on the figures by thin grey lines and noted by matching symbol type for each timeseries. In iv) the model was not fit to successfully fed data so has no overlaid points. Anopheles gambiae s.l. (circles), An. funestus s.l. (squares) and An. arabiensis (triangles) mosquitoes are noted. The data from the comprehensive source are only used in aggregated format.

Fig 4: Model predictions of proportion of mosquitos killed and successfully fed for the best-estimate (median posterior predictive value) across all study datasets, 50% (darker shaded region) and 95% (lighter shaded region) credible intervals (CrI) from the respective fits to the comprehensive data. Subfigures i) and ii) show results using comprehensive data only (Model 2b), and subfigures iii) and iv) show results using all datasets. Data for the fits are overlaid onto subfigures i) and ii) to demonstrate the suitability of the time-dependent functions. In these fits, the individual study predictions are overlaid on the figures by thin grey lines and noted by matching symbol type for each timeseries. As Model 3 is inferred from both AD and CD sources, there are no data to overlay.

Fig 5: A forest plot of posterior predictions for all models considered, showing median, 25% and 75% percentiles for dead and survived fed probabilities across all study datasets at 3, 6 and 12 months. Individual study predictions are shown with grey points.

6) Line 248-251: “The model using comprehensive data only is very uncertain. However, these data provide worthwhile additional information when combined with the aggregate data such that the proportion survived and fed at 12 months could be approximately 10% higher than the aggregate data only model..”

This is eluded to in the text below but this uncertainty is likely due to the small amount comprehensive data and the large amount of aggregate data in comparison, rather than directly due to the models themselves?

Perhaps it would be better to describe that the results or estimates using comprehensive data only are uncertain, rather than the models themselves?

We completely agree with the reviewer. This was the original intention. We have updated the text as suggested.

Lines 273-274: “The model results using comprehensive data only are very uncertain, largely due to few time series available.”

7) A general query, related somewhat to the discussion. The series of models here are clearly suitable for addressing the particular question in hand related to mosquito feeding but are these models ‘tailor made’ only for this question? Are there any features of the models which could be generalizable to a wider set of research questions or contexts? Bringing this out a little more in the discussion may be helpful

Thank you for the question. These models are not tailor-made as such but can and have been adopted elsewhere. In fact, the reverse is true, meaning we wanted to see if models that have been successfully used elsewhere for similar problems could be used in this context. We have added the following text to the discussion:

Lines 344-349: “The approach detailed here can be thought of generally as a method of model calibration using indirect data. In our case, this is using marginal (aggregate) and joint (comprehensive) data to inform a joint distribution, but the problem can be more general than this and the data less directly relevant. The approach provides a principled framework to synthesise multiple, different data sets. To date, Bayesian evidence synthesis has been applied to a range of problems, for example, to estimating HIV and Hepatitis C prevalence.”

---

## [Decision Letter · Decision Letter 1]

20 Jan 2022

An evidence synthesis approach for combining different data sources illustrated using entomological efficacy of insecticides for indoor residual spraying

PONE-D-21-15152R1

Dear Dr. Green,

Thank you for your patience! The external reviewer and myself have reviewed your revised manuscript. Congratulations! The manuscript has been judged scientifically suitable for publication and will be formally accepted for publication once it meets all outstanding technical requirements.

Kind regards,

Monika Gulia-Nuss, PhD

Academic Editor

PLOS ONE

Additional Editor Comments (optional):

Reviewers' comments:

Reviewer's Responses to Questions

**Comments to the Author**

1. If the authors have adequately addressed your comments raised in a previous round of review and you feel that this manuscript is now acceptable for publication, you may indicate that here to bypass the “Comments to the Author” section, enter your conflict of interest statement in the “Confidential to Editor” section, and submit your "Accept" recommendation.

Reviewer #1: All comments have been addressed

2. Is the manuscript technically sound, and do the data support the conclusions?

Reviewer #1: Yes

3. Has the statistical analysis been performed appropriately and rigorously? 

Reviewer #1: Yes

4. Have the authors made all data underlying the findings in their manuscript fully available?

Reviewer #1: Yes

5. Is the manuscript presented in an intelligible fashion and written in standard English?

Reviewer #1: Yes

6. Review Comments to the Author

Reviewer #1: Thank you for addressing all concerns raised. The manuscript is now recommended for publication. *****

7. PLOS authors have the option to publish the peer review history of their article (what does this mean?). If published, this will include your full peer review and any attached files.

Reviewer #1: **Yes: **Koen Dechering

---

## [Editor Report · Acceptance letter]

15 Mar 2022

PONE-D-21-15152R1 

An evidence synthesis approach for combining different data sources illustrated using entomological efficacy of insecticides for indoor residual spraying 

Dear Dr. Green:

I'm pleased to inform you that your manuscript has been deemed suitable for publication in PLOS ONE. Congratulations! Your manuscript is now with our production department. 

Kind regards, 

on behalf of

Dr. Monika Gulia-Nuss 

Academic Editor

PLOS ONE